# Strong and selective isotope effect in the vacuum ultraviolet photodissociation branching ratios of carbon monoxide

Pan Jiang[1,2], Xiaoping Chi[1,2], Qihe Zhu[1], Min Cheng [1] & Hong Gao [1]

Rare isotope ([13]C, [17]O and [18]O) substitutions can substantially change absorption line positions, oscillator strengths and photodissociation rates of carbon monoxide (CO) in the vacuum ultraviolet (VUV) region, which has been well accounted for in recent photochemical models for understanding the large isotopic fractionation effects that are apparent in carbon and oxygen in the solar system and molecular clouds. Here, we demonstrate a strong isotope effect associated with the VUV photodissociation of CO by measuring the branching ratios of $^{12}C^{16}O$ and $^{13}C^{16}O$ in the Rydberg 4p(2), 5p(0) and 5s(0) complex region. The measurements show that the quantum yields of electronically excited C atoms in the photodissociation of $^{13}C^{16}O$ are dramatically different from those of $^{12}C^{16}O$, revealing strong isotope effect. This isotope effect strongly depends on specific quantum states of CO being excited, which implies that such effect must be considered in the photochemical models on a state by state basis.

[1] Beijing National Laboratory for Molecular Sciences (BNLMS), Institute of Chemistry, Chinese Academy of Sciences, Beijing 100190, China. [2] University of Chinese Academy of Sciences, Beijing 100049, China. Correspondence and requests for materials should be addressed to M.C. (email: chengmin@iccas.ac.cn) or to H.G. (email: honggao2017@iccas.ac.cn)

Carbon monoxide (CO) is one of the most important molecules in astronomy, not only because it is the second most abundant molecular species in the universe only after the molecular hydrogen (H$_2$) and the main gas phase reservoir of interstellar carbon and oxygen atoms[1–3], but also because it plays a central role in understanding the large isotopic fractionation effects which are apparent in carbon and oxygen in the solar system and molecular clouds due to its intrinsic photochemical properties[4,5]. It has long been known that rare isotope ($^{13}$C, $^{17}$O, and $^{18}$O) substitutions can substantially change photoabsorption line positions, oscillator strengths and photodissociation rates of CO[5–8], and these are key input parameters for the self-shielding models[9–11]. These models have been proposed for explaining the anomalous oxygen isotopic distributions observed among various early Solar System objects which contain both $^{16}$O-rich and $^{16}$O-poor reservoirs[12]. Self-shielding process happens when a light beam with broad spectral distribution penetrates a sample of mixed gaseous molecules which have different absorption line positions. Due to the different column densities of the gaseous components, which result in different attenuation speeds according to Beer's law, the relative amount of light absorption for different molecular species will change along the light propagation direction. To test the self-shielding models, Thiemens and co-workers performed an experiment at the advanced light source in Berkeley, strong and wavelength selective mass-independent oxygen isotope fractionation processes have been observed, from which they concluded that the self-shielding effect is not the major reason for the anomalously enriched atomic oxygen reservoirs observed in the Solar System[13]. However, their experiment and discussion did not convince most of the other researchers[14–16]. Now it is generally agreed that self-shielding effect is the main reason of the massive isotope fractionation observed in the above experiment and also that in the solar system, while Thiemens' experiment does have the effect of encouraging people to include the different photoabsorption and photodissociation cross sections for different CO isotopologues in their models, which has been proved to make the modeling results in closer agreement with the experimental measurements[17,18]. Compared to the absorption line positions, accurate absolute values of photoabsorption and photodissociation cross sections are much more difficult to be measured, thus continued recent efforts are being devoted to obtain more precise values of these for all the CO isotopologues in hope of further improving the photochemical modeling[19–22].

While most of the current photochemical models have considered the rare isotope substitution effects in the photodestruction process of CO due to VUV absorption, only few of them have talked about what chemical processes would happen following the production of C and O atoms, which can play important roles in sequestering the isotope fractionation results generated in the photodissociation process[23–25]. Recent measurements of the solar wind material collected by the Genesis mission showed that both heavy isotopes of oxygen ($^{17}$O and $^{18}$O) are depleted in the Sun compared with that in the Earth and other terrestrial planetary materials by ~7%[26], and the heavy nitrogen isotope ($^{15}$N) is depleted even more in the Sun by ~40%[27]. The reasons for such big difference between O and N in the degree of heavy isotope depletion in the Sun are still unclear. Clayton proposed that this is caused by the different reactivity of the atomic photodissociation products of CO and N$_2$ when react with H$_2$[23]. He assumed that dissociation of CO only produces O atoms in the ground state O ($^3$P), and that of N$_2$ can produce 50% N atoms in the excited state N($^2$D). N($^2$D) can be much more efficiently trapped due to its higher reaction rates with H$_2$ compared with O($^3$P), and this caused the much higher degree of depletion for $^{15}$N compared with $^{17}$O and $^{18}$O in the Sun as observed by the Genesis mission. To verify this assumption, the photodissociation branching ratios

of $^{12}$C$^{16}$O have been measured recently[28–33], which showed that photodissociation of $^{12}$C$^{16}$O not only generates C and O atoms in the ground state, but also produces significant amount of C and O atoms in the excited states, and the ratios between them strongly depend on the specific rovibronic state of $^{12}$C$^{16}$O being excited. These observations clearly showed that the initial assumption by Clayton is oversimplified. By using the measured branching ratios and the photoabsorption cross sections of $^{12}$C$^{16}$O and $^{14}$N$_2$ reported in the literature, Shi et al. carefully modeled the possible fractionation processes of O and N isotopes in the framework of self-shielding models by considering the different chemical reactivity between the excited O($^1$D) [N($^2$D)] and the ground O ($^3$P) [N($^4$S)] produced by VUV photodissociation of CO [N2]. The modeling results qualitatively support the observations by the Genesis mission[24]. Besides O and N, Lyons et al. have recently showed that the heavy C isotope ($^{13}$C) is also depleted in the Sun compared with that in the Earth and other planetary objects[25]. In their work, they have also considered the possible effect caused by the production of excited C atoms C($^1$D) from photodissociation of CO, because C($^1$D) can react with H$_2$ several orders of magnitude faster than any other reactions in their models.

Even though not mentioned in the above two examples, they actually both assumed that all CO isotopologues produce the same amount of O and C atoms in the excited states, the validity of which has not been experimentally proved yet[24,25]. In the present work, we measure the photodissociation branching ratios of $^{13}$C$^{16}$O in the Rydberg 4p(2), 5p(0) and 5s(0) complex region and find that the percentages of producing C($^1$D) and O($^1$D) for $^{13}$C$^{16}$O can be dramatically different from that for $^{12}$C$^{16}$O in the same absorption band, revealing very strong isotope effect. The underlying mechanism for such isotope effect is general for any indirect predissociation processes[34], like that in O$_2$, N$_2$. This could potentially have great impact on the current photochemical models for understanding the various isotope fractionation phenomena observed in the fields of astrochemistry, geochemistry, and planetary atmospheric chemistry[35,36].

## Results

**Isotope dependent photodissociation branching ratios.** The present photodissociation branching ratio measurements were performed on a time-slice velocity-map ion imaging apparatus (TSVMI) (Fig. 1c) which equips with a high-resolution tunable VUV laser radiation source (spectral linewidth ~ 0.3 cm$^{-1}$) generated by the two-photon resonance-enhanced four-wave mixing scheme (Fig. 1d), as shown schematically in Fig. 1 (see "Methods" section for more details)[32,37]. A pulsed supersonic molecular beam of CO ($^{12}$C$^{16}$O or $^{13}$C$^{16}$O) was crossed by the VUV laser beam in the photodissociation and photoionization (PD/PI) region of the TSVMI setup. CO molecules absorbed a single VUV photon to be excited to a specific rovibronic state, and then predissociated into the three possible channels as follows (see Fig. 1a, b):

$$CO(X^1\Sigma^+) + h\nu_{VUV} \rightarrow C(^3P) + O(^3P) \quad h\nu_{VUV} \geq 11.09 eV \quad (1)$$

$$C(^1D) + O(^3P) \quad h\nu_{VUV} \geq 12.35 eV \quad (2)$$

$$C(^3P) + O(^1D) \quad h\nu_{VUV} \geq 13.06 eV \quad (3)$$

The TSVMI setup collected the velocity-map ion images of C atoms produced in the above photodissociation processes. The images were then converted to the total kinetic energy release (TKER) spectra, from which the branching ratios of the above three dissociation channels can be deduced (see "Methods" section).

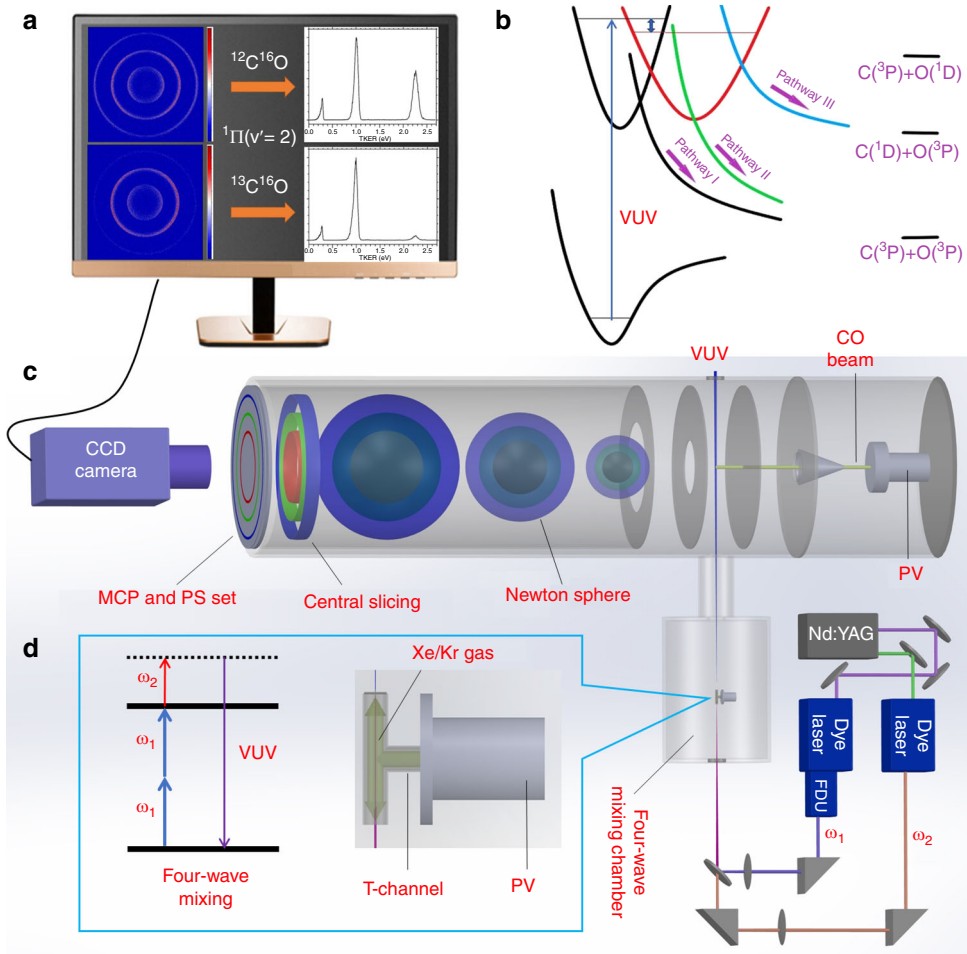

**Fig. 1** Schematic diagram of the experimental setup. **a** Raw time-slice velocity map imaging (TSVMI) images and the corresponding total kinetic energy release (TKER) spectra for $^{12}C^{16}O$ and $^{13}C^{16}O$ in the absorption band with upper state of $^{1}\Pi(v' = 2)$. **b** Three possible dissociation pathways of CO following the absorption of a single VUV photon: Pathways I and II go to $C(^{3}P)+O(^{3}P)$, Pathway III goes to $C(^{1}D)+O(^{3}P)$ (see text for details). **c** Time-slice velocity-map ion imaging system. **d** Tunable VUV laser radiation source generated by the two-photon resonance-enhanced four-wave mixing scheme using a Xe or Kr gas jet as the nonlinear medium. MCP micro-channel plate; PS phosphorscreen; VUV vacuum ultraviolet; PV pulsed valve; FDU frequency doubling unit

In the present study, we measured the branching ratios of the above three channels for the photodissociation of $^{13}C^{16}O$, and compared with those of $^{12}C^{16}O$ from previous studies in the Rydberg 4p(2), 5p(0) and 5s(0) complex region (92.85–94.00 nm), where complicated photoabsorption structures due to various Rydberg–Rydberg and Rydberg–valence interactions have been observed, and the absorption line positions and oscillator strengths were found to strongly depend on the isotopic substitution[8,19,20]. The measured branching ratios of seven absorption bands in the wavelength range 92.85–94.00 nm are listed in Table 1, and selected four pairs of raw images and the corresponding TKER spectra are compared in Fig. 2 for the photodissociation of $^{12}C^{16}O$ and $^{13}C^{16}O$. The branching ratio data for absorption bands of $^{1}\Pi(v' = 2)$, $(4p\pi)\,^{1}\Pi(v' = 2)$, $(5p\pi)\,^{1}\Pi(v' = 0)$, $(5p\sigma)\,^{1}\Sigma^{+}(v' = 0)$ and I $^{1}\Pi$ for $^{12}C^{16}O$ are adopted from our previous study[32], and the newly measured branching ratios in the current study for the absorption bands of $C'^{1}\Sigma^{+}(v' = 7)$ and I′(5s$\sigma$) $^{1}\Sigma^{+}(v' = 0)$ for $^{12}C^{16}O$ are consistent with the previous measurement in the literature[31]. In most of the raw TSVMI images as shown in Fig. 2, three rings are observed, from the outermost ring to the innermost, which are corresponding to the three dissociation channels $C(^{3}P)+O(^{3}P)$, $C(^{1}D)+O(^{3}P)$ and $C(^{3}P)+O(^{1}D)$, respectively. The outermost, middle and

innermost rings in the TSVMI images are converted to the peaks with largest (2.0–2.3 eV), medium (0.8–1.0 eV) and smallest (0.1–0.3 eV) kinetic energies in the TKER spectra, respectively. After normalization by using the photoionization cross sections of $C(^{3}P)$ and $C(^{1}D)$ (see Methods)[38,39], the relative intensities of the three peaks in the TKER spectra directly reveal the branching ratios of the three dissociation channels.

In Fig. 2, we can see that the relative intensities of the three peaks in the TKER spectra, and thus the branching ratios are dramatically different between $^{12}C^{16}O$ and $^{13}C^{16}O$ in the absorption bands with upper states $^{1}\Pi(v' = 2)$, $(5p\pi)\,^{1}\Pi(v' = 0)$, $C'^{1}\Sigma^{+}(v' = 7)$ and I′(5s$\sigma$) $^{1}\Sigma^{+}(v' = 0)$. This is especially prominent for the comparison between the lowest dissociation channel $C(^{3}P)+O(^{3}P)$ and the spin-forbidden channel $C(^{1}D)+O(^{3}P)$, which respectively produce C atoms in the ground and first excited states. The largest variations happen in the absorption bands of $^{1}\Pi(v' = 2)$ and $C'^{1}\Sigma^{+}(v' = 7)$, where the relative ratios of $C(^{3}P)+O(^{3}P)$ and $C(^{1}D)+O(^{3}P)$ are completely inverted. For $^{12}C^{16}O$, dissociation into the lowest channel dominates the predissociation process with percentages of ~53% and ~77% for the absorption bands of $^{1}\Pi(v' = 2)$ and $C'^{1}\Sigma^{+}(v' = 7)$, respectively; while for $^{13}C^{16}O$, dissociation into the spin-forbidden channel producing C atoms in the excited state dominates the

**Table 1 Measured photodissociation branching ratios of $^{12}C^{16}O$ and $^{13}C^{16}O$**

| Upper state | VUV(cm$^{-1}$) | | | [C($^3$P)+O($^1$D)]: [C($^1$D)+O($^3$P)]: [C($^3$P)+O($^3$P)]$^e$ | |
|---|---|---|---|---|---|
| | $^{12}C^{16}O$ | $^{13}C^{16}O^a$ | $\Delta\nu^b$ | $^{12}C^{16}O$ | $^{13}C^{16}O$ |
| $^1\Pi(v'=2)$ | 107685.8, R(0) | 107596.6, R(0) | 89.2 | [9.5 ± 1.8]:[37.8 ± 4.0]:[52.6 ± 5.8]$^c$ | [12.5 ± 0.5]:[78.9 ± 1.1]:[8.6 ± 0.7] |
| $(4p\pi)$ $^1\Pi(v'=2)$ | 107523.5, R(0) | 107431.7, R(0) | 100.8 | [11.9 ± 2.0]:[60.2 ± 3.0]:[27.9 ± 4.7]$^c$ | [10.5 ± 0.2]:[51.5 ± 0.5]:[38.0 ± 0.3] |
| $(5p\pi)$ $^1\Pi(v'=0)$ | 107336.8, R | 107291.4, R(0) | 45.4 | [15.7 ± 0.1]:[22.0 ± 0.3]:[62.3 ± 0.3]$^c$ | [6.7 ± 0.4]:[48.1 ± 1.0]:[45.2 ± 1.3] |
| $(5p\sigma)$ $^1\Sigma^+(v'=0)$ | 107220.0$^f$ | 107201.3$^f$ | 18.7 | [6.6 ± 0.8]:[12.3 ± 0.9]:[81.1 ± 1.6]$^c$ | [6.3 ± 0.6]:[25.3 ± 1.4]:[68.4 ± 1.9] |
| I $^1\Pi$ | 107159.0 | 107110.9 | 48.1 | [2.0 ± 0.2]:[14.2 ± 1.0]:[83.8 ± 1.3]$^c$ | [0.5 ± 0.05]:[6.5 ± 0.2]:[93.0 ± 0.2] |
| C$'$ $^1\Sigma^+(v'=7)^d$ | 106882.8 | 106790.2 | 92.6 | [3.0 ± 0.3]:[19.7 ± 0.6]:[77.2 ± 0.3] | [16.3 ± 0.5]:[61.7 ± 0.6]:[22.0 ± 0.3] |
| I$'$(5s$\sigma$) $^1\Sigma^+(v'=0)$ | 106400.3 | 106362.1, R(3) | 38.2 | [0]:[0]:[100] | [0.7 ± 0.1]:[2.4 ± 0.1]:[96.9 ± 0.1] |

$^a$The VUV photon energies are calibrated according to the Rydberg series of Xe: $5p^5(^2P^o_{1/2})ns$ ($n=12$–17), the uncertainty should be within 1.5 cm$^{-1}$
$^b$The VUV photon energy differences between $^{12}C^{16}O$ and $^{13}C^{16}O$, which approximately represent the energy shifts of the corresponding rovibronic states caused by the isotopic substitution of $^{12}C$ by $^{13}C$
$^c$The branching ratios are adopted from ref. [32]
$^d$The assignment of this state is according to ref. [8]. It is usually labeled as $^1\Sigma^+(v'=2)$ in the community of astrophysics, for example ref. [17]
$^e$The error bars of the presented branching ratios are the standard deviations (1$\sigma$) of three independent experimental trials, which has not considered the possible uncertainties of the photoionization cross sections of C($^3$P) and C($^1$D) in ref. [38,39], see "Methods" section
$^f$This corresponds to the position of the band head

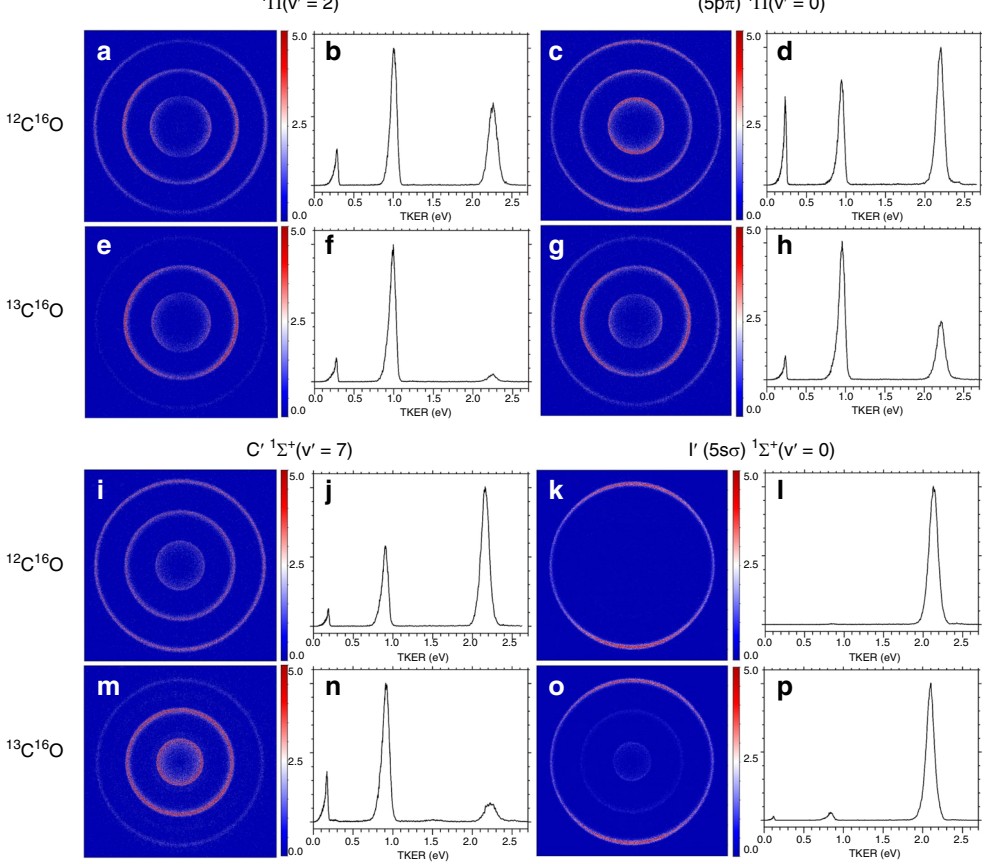

**Fig. 2** State dependent isotope effect of the photodissociation branching ratios. The comparisons of the raw time-slice velocity map imaging (TSVMI) images and the corresponding total kinetic energy release (TKER) spectra between the photodissociations of $^{12}C^{16}O$ and $^{13}C^{16}O$ for the absorption bands with upper states of $^1\Pi(v'=2)$ (**a**, **b**, **e**, **f**), (5p$\pi$) $^1\Pi(v'=0)$ (**c**, **d**, **g**, **h**), C$'^1\Sigma^+(v'=7)$ (**i**, **j**, **m**, **n**) and I$'$(5s$\sigma$) $^1\Sigma^+(v'=0)$ (**k**, **l**, **o**, **p**). The outermost, middle and innermost rings in the TSVMI images are corresponding to the peaks with largest (2.0–2.3 eV), medium (0.8–1.0 eV) and smallest (0.1–0.3 eV) kinetic energies in the TKER spectra, respectively. The heights of the three peaks are rescaled by setting the highest peak in each spectra to 1. The underneath areas of the peaks represent the relative intensities (or branching ratios) of the three photodissociation channels C($^3$P)+O($^3$P), C($^1$D)+O($^3$P) and C($^3$P)+O($^1$D) respectively after normalization by using the photoionization cross sections of C($^3$P) and C($^1$D) (see Methods)

predissociation process with percentages of ~79% and ~62%, and the corresponding percentages of the lowest dissociation channel decrease to ~9% and ~22%, respectively for the two absorption bands. These huge branching ratio variations (up to 50%) between $^{12}C^{16}O$ and $^{13}C^{16}O$ in the above two absorption bands have been

directly revealed in Fig. 2a, b, e, f, i, j, m, n, respectively. Relatively large variation of the branching ratio has also been observed for the absorption band of (5p$\pi$) $^1\Pi(v'=0)$, in which case the substitution of $^{12}C$ by $^{13}C$ decreases the relative percentage of C($^3$P)+O($^3$P) from ~62% to ~45%, and increases that of

$C(^1D)+O(^3P)$ from ~22% to ~48%, as shown in Fig. 2c, d, g, h. In the case of the absorption band with upper state $I'(5s\sigma)$ $^1\Sigma^+(v' = 0)$, only the outermost ring corresponding to the lowest dissociation channel can be observed for $^{12}C^{16}O$, as shown in Fig. 2k, l, which indicates that dissociation into the two spin-forbidden channels generating excited C and O atoms is negligibly weak in this absorption band for $^{12}C^{16}O$. While for $^{13}C^{16}O$, the two inner rings and the corresponding peaks are clearly observable from the TSVMI image and the TKER spectrum, as shown in Fig. 2o, p. Besides the above four absorption bands, the substitution of $^{12}C$ by $^{13}C$ has also enhanced the generation of excited C atoms in the absorption band of $(5p\sigma)$ $^1\Sigma^+(v' = 0)$ at the band head, i.e. from ~ 12% to ~25%. Other than enhancing the generation of $C(^1D)$, $^{13}C$ substitution could also decrease it as in the cases of $(4p\pi)$ $^1\Pi(v' = 2)$ and $I$ $^1\Pi$, while the decreased amount is relatively small, less than 10% for both of the two absorption bands, as shown in Table 1. Compared with the large variations of the channel $C(^1D)+O(^3P)$ caused by the $^{13}C$ substitution, relatively small changes on the percentages of the channel $C(^3P)+O(^1D)$ have been observed in the energy range investigated in the current study, which are usually less than 10%. As shown in Fig. 2c, d, g, h, i, j, m, n, the largest variations are observed in the absorption bands of $(5p\pi)$ $^1\Pi(v' = 0)$ and C $'^1\Sigma^+(v' = 7)$, in which the percentages of generating excited O atoms decreases and increases by ~10%, respectively.

From the above presentation, we can see that the substitution of $^{12}C$ by $^{13}C$ can greatly change the photodissociation branching ratios of CO, which varies from less than ~10% to ~50%. This isotopic effect strongly depends on the specific electronic and vibrational quantum level of CO being excited. Previous studies have shown that the photodissociation branching ratios of CO also depend on the rotational quantum number of the excited state, which revealed the detailed photodissociation dynamics[30,32]. In the current study, we investigate the rotational dependence of the branching ratio of the absorption band with upper state $^1\Pi(v' = 2)$ for $^{13}C^{16}O$, and compare it with that of $^{12}C^{16}O$ in Fig. 3. This measurement shows that the huge branching ratio variation due to the isotopic substitution is not restricted to the single rotational level that is presented above for the absorption band of $^1\Pi(v' = 2)$. Our previous study revealed that the branching ratios of the lowest dissociation channel $C(^3P)+O(^3P)$ increase linearly as a function of $J'(J'+1)$ with $J'$ being the rotational quantum number of the upper state[32]. The linear dependence is due to the heterogeneous perturbations by the repulsive $D'^1\Sigma^+$ state that is correlating to the dissociation channel $C(^3P)+O(^3P)$. As shown in Fig. 3, the pattern of the rotational dependence for $^{13}C^{16}O$ is almost the same with that of $^{12}C^{16}O$, the substitution of $^{12}C$ by $^{13}C$ shifts up the line representing the branching ratios of the channel $C(^1D)+O(^3P)$, and at the same time shifts down the line for $C(^3P)+O(^3P)$, without changing the slopes of the lines. This observation implies that the isotopic substitution has negligible effect on the electronic perturbation coefficient between the Rydberg state $^1\Pi(v' = 2)$ and the repulsive $D'^1\Sigma^+$ state, while it does greatly enhance the coupling schemes which finally dissociate CO molecules into the spin forbidden (or triplet) channels. This will be discussed furthermore in next section.

**Origins of the isotope effect**. In the above section, it is shown that the photodissociation branching ratios of $^{13}C^{16}O$ are dramatically different from that of $^{12}C^{16}O$, revealing conspicuous isotope effect which depends on the specific quantum state of CO being excited. To our knowledge, this isotope effect has not been noticed before. From the chemical physics point of view, the substitution of $^{12}C$ by $^{13}C$ must have dramatically changed the relative coupling strengths among different rovibronic states

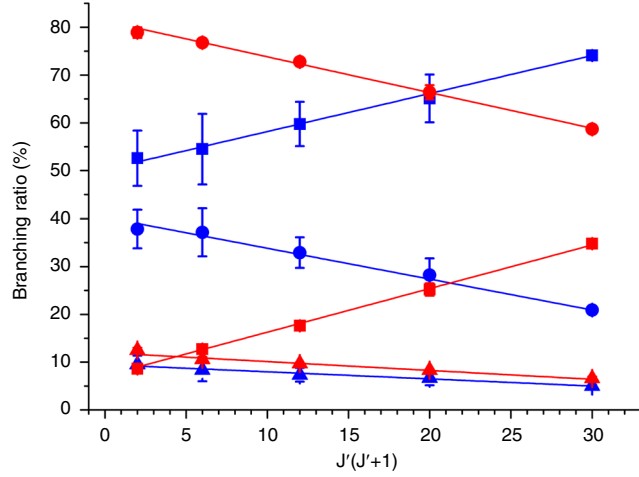

**Fig. 3** Rotational dependence of the photodissociation branching ratios. Dependence of the branching ratios on $J'(J'+1)$ for the $^1\Pi(v' = 2)$ state of $^{12}C^{16}O$ (blue) and $^{13}C^{16}O$ (red) measured from the R-branch, $J'$ is the rotational quantum number of the upper level. Square: $C(^3P)+O(^3P)$; Circle: $C(^1D)+O(^3P)$; Triangle: $C(^3P)+O(^1D)$. The data of $^{12}C^{16}O$ are adopted from ref. [32]. The error bars represent the standard deviation ($1\sigma$) of three independent measurements

which determine the partitions of the CO photodissociation into the three available channels. Due to the complicated potential curves of CO and the mutual interactions among them in this high excited region[40], quantitative explanations of the observed isotope effect is obviously out of scope for the current study. Here we employ a simple diabatic coupling model to qualitatively explain the strong isotope effect observed in the current study. This model was recently discussed by Thiemens and coworkers for explaining the variations of absorption line positions and oscillator strengths of $N_2$ due to isotopic substitution, which should also work for the photoabsorption and photodissociation processes of CO as pointed out by those authors[34,41]. This is actually similar to the coupled equations method used by Lefebvre-Brion and coworkers for quantitatively predicting the line positions, oscillator strengths, accidental perturbations, and photodissociation branching ratios of CO and their dependences on the isotopic substitution[40,42,43]. In this model, two bound diabatic potential curves with different bonding types, usually one Rydberg and one valence, cross at certain point and interact with each other through an electronic coupling constant $H$, this is shown schematically in Fig. 1b between the directly excited Rydberg state (the bound black curve) and a nearby valence state (the red curve). The specific coupling strength between two discrete quantum levels from the two diabatic potential curves is scaled by the energy gap of the two levels as $H/(E_a-E_b)$, where $E_a$ and $E_b$ are the energy positions of the two interacting quantum states as shown by the double head arrow in Fig. 1b. According to Born–Oppenheimer approximation, isotopic substitution should not dramatically change the electronic coupling constant $H$, but it does modify the reduced masses of diatomic molecules, this will, in turn, change the vibrational and rotational constants of the molecules, and thus the energy positions $E_a$ and $E_b$. Because the two diabatic potential curves have different bonding characters (Rydberg vs valence), variations of the reduced mass change their vibrational and rotational energy ladders in different ways, this will modify the energy gap $E_a-E_b$ between the two interacting levels, and thus the coupling strength between them (Please refer to Fig. 1 of ref. [34] for an example). This mechanism has been argued to be the origin of the strong isotopic selectivity in absorption line positions and oscillator strengths of $N_2$, especially

in the crossing region of the two diabatic potential curves[34]. This physical picture can also explain the isotopically selective accidental local perturbations which are ubiquitous in the absorption spectra of $N_2$, CO, and their isotopologues[42,44].

Table 1 lists the energy shifts caused by the isotopic substitution for all the directly excited electro-vibrational states investigated in the current study. It shows large differences from state to state varying from ~19 cm$^{-1}$ for the (5p$\sigma$) $^1\Sigma^+$($v' = 0$) state to ~101 cm$^{-1}$ for the (4p$\pi$) $^1\Pi$($v' = 2$) state, this observation is consistent with the above discussion that isotopic substitution shifts the vibrational energy ladders in different manners for electronic states of different bonding natures. It has long been known that photodissociation of CO is an indirect process with all the possible dissociation pathways summarized before[30,45]. There are two different pathways for dissociating into the lowest channel C($^3$P)+O($^3$P), Pathways I and II, as shown schematically in Fig. 1b. Pathway I is through coupling with the well-known D$'^1\Sigma^+$ state (the repulsive black curve), which leads to the linear dependence of the photodissociation rates and branching ratios on $J'(J'+1)$ for states of $^1\Pi$ symmetry, like the $^1\Pi$($v' = 2$) state as shown in Fig. 3. The D$'^1\Sigma^+$ state is repulsive in the Franck-Condon region, thus the above physical model, which is for interaction between two bound electronic states, does not work[34]. The coupling strength between a bound electro-vibrational level and a repulsive state will not be significantly changed by the small energy shift of the bound level caused by isotopic substitution if it is well separated from the curve crossing region, this might explain why the substitution of $^{12}$C by $^{13}$C did not change the way how the branching ratios depend on $J'$($J'+1$) for the $^1\Pi$($v' = 2$) state, i.e. the slopes of the linear fitting lines are the same for both $^{12}$C$^{16}$O and $^{13}$C$^{16}$O, as shown in Fig. 3. Pathway II is to first couple with another bound state of $^1\Pi$ or $^3\Pi$ symmetry (the red curve), then dissociate into the channel C($^3$P)+O($^3$P) through a repulsive state of $^1\Pi$ or $^3\Pi$ symmetry (the green curve). Even though the detailed dynamics of this pathway has not been clarified yet, its existence has been confirmed in several previous studies[30,41]. Because the first step involves coupling between two bound states, its strength could be strongly affected (increased or decreased) by isotopic substitution as discussed above. Reduction of the total contribution from Pathway II could be one of the reasons why the relative percentage into the C($^3$P)+O($^3$P) channel is greatly reduced for $^{13}$C$^{16}$O compared with $^{12}$C$^{16}$O in the $^1\Pi$($v' = 2$) state. Besides the above discussed reason, it could also be due to the enhancement of the photodissociation rate into the channel C($^1$D)+O($^3$P) which, in turn, decreases the relative percentage of the channel C($^3$P)+O($^3$P). Since the directly accessible quantum state from the ground electronic state by single VUV photon absorption is singlet, which cannot directly correlate with the channel C($^1$D)+O($^3$P), it usually couples with a repulsive state of $^3\Pi$ symmetry through spin-orbit interaction to dissociate into C($^1$D)+O($^3$P). In this process, coupling with an intermediate bound state of $^1\Pi$ or $^3\Pi$ symmetry (the red curve) as a first step could be possible[30,41]. This mechanism is shown in Fig. 1b as Pathway III. The coupling strength between the directly excited and the intermediate states can be very sensitive to the isotopic substitution as discussed above, and this could be the second reason for the great enhancement of the C($^1$D)+O($^3$P) channel for $^{13}$C$^{16}$O compared with $^{12}$C$^{16}$O in the $^1\Pi$($v' = 2$) state. Dissociation pathways into the channel C($^3$P)+O($^1$D) are similar to that into the C($^1$D)+O($^3$P) channel, thus will not be repeated here.

Thus except for Pathway I which involves the repulsive D$'^1\Sigma^+$ state, the coupling strengths in the other two pathways as summarized in the previous study[30] can be both significantly altered by the isotopic substitution, and this should be the origins

of the strong isotope effect of the photodissociation branching ratios of CO observed in the current study. One thing to be noticed is that the magnitude of the isotope effect does not depend on the amount of energy shift in a monotone way. For example, the biggest energy shift (~101 cm$^{-1}$) is observed for the (4p$\pi$) $^1\Pi$($v' = 2$) state, while its branching ratios have only been changed by ~10%; on the other hand, the energy shift for the (5p$\pi$) $^1\Pi$($v' = 0$) state is only ~45 cm$^{-1}$, its branching ratios are varied by ~25% due to the isotopic substitution, as shown in Table 1. This is because of the fact that the coupling strength depends on the energy difference of the two states, $E_a - E_b$, which not only depends on the energy position of the directly excited state ($E_a$), but also on the energy position of the state that is being coupled with ($E_b$). Thus, it is not possible to tell directly from the isotopically selective photoabsorption spectra that how much the branching ratio of a specific absorption band will be changed by the isotopic substitution. Quantitative experimental measurement like that presented in the current study is needed for all the CO absorption bands, and this is being performed in our lab.

## Discussion

Early photochemical models based on the self-shielding effect only considered the isotopic fractionation effect caused by the shifts of photoabsorption line positions due to isotopic substitutions and assumed that no other isotopic fractionation processes occurred in the photodissociation process. This assumption was directly challenged recently by Thiemens and coworkers who found that the relative degrees of enrichment between $^{17}$O and $^{18}$O are mostly not unity and strongly wavelength dependent. This contradicts with the prediction by photochemical models which only consider the self-shielding effect[13]. Being stimulated by this experimental work, Lyons performed a detailed quantitative simulation, and found that pure self-shielding alone can account for the magnitude of the massive isotopic fractionation observed in the experiment, while consideration of other isotopic effects can definitely make the model more realistic, for example, the isotopic differences in dissociation probabilities can cause the slope increase observed in Thiemens' experiment[18]. Even though the branching ratios of $^{12}$C$^{17}$O and $^{12}$C$^{18}$O have not been directly measured yet, the present experiment strongly implies that different relative amounts of excited $^{17,18}$O atoms [O($^1$D)] might be possibly produced, especially when considering the fact that $^{13}$C$^{16}$O and $^{12}$C$^{18}$O have very similar reduced masses[35]. This could affect the subsequent trapping reaction rate between CO and O which forms $CO_2$ for detection in Thiemens' experiment[18]. Since CO photodissociation can only generate excited O atoms [O($^1$D)] at VUV wavelength shorter than 94.94 nm (see reaction (3)), thus the present isotope effect observed for the branching ratio should not affect any absorption bands at longer wavelength in Thiemens' experiment. Direct measurements of the photodissociation branching ratios for $^{12}$C$^{17}$O and $^{12}$C$^{18}$O have been planned in our lab, since they might be important for any photochemical models based on the self-shielding effect, for example, the recent one by Shi et al. for explaining why the rare isotope $^{15}$N is much more enriched (~7 times) than the rare isotopes $^{17}$O and $^{18}$O in the terrestrial planets relative to the Sun as revealed by the recent NASA's Genesis mission[24,26,27].

Besides the O isotopes, C isotopes are also of central importance for understanding the formation history of the Solar system. By reanalyzing the ATMOS Fourier transform spectrometer data for CO[46], Lyons et al. concluded that the primary reservoirs of C on the terrestrial planets are enriched in $^{13}$C relative to the Sun, similar to O and N isotopes[25]. To explain this finding, they have built a photochemical timescale model which included all the relevant

chemical reactions between C atoms and other molecular species, like $H_2$, CH, $CH_2$, and also VUV photons, that could be present in the early Solar system or parent molecular cloud. In the model, they have highlighted the potential possible roles played by the excited $C(^1D)$ atoms produced from the photodissociation of CO, because the reaction of $C(^1D)$ towards $H_2$ is faster by a factor of ~$10^3$ than any other photochemical processes in the model, this could sequester the C atoms into large molecules to at least partially preserve the $^{13}C$ isotope signature formed in the self-shielding process before they are ionized by VUV photons. The photo-ionization process can erase the $^{13}C$ isotope signature through the ion exchange reaction between $C^+$ and CO. Our current study showed that at some absorption bands of $^{13}C^{16}O$, the production of excited $C(^1D)$ atoms can be significantly enhanced compared with that in $^{12}C^{16}O$, for example, at the $^1\Pi(v'=2)$ and $C'^1\Sigma^+(v'=7)$ bands the relative percentages of $C(^1D)$ are increased by ~50% as shown in Fig. 2 and Table 1. This could potentially make the enrichment of $^{13}C$ more preferential than $^{12}C$ due to the much faster reaction rates of $C(^1D)$ than that of the ground state $C(^3P)$ when react with $H_2$. Without systematic branching ratio measurements like that in the present study for all the relevant $^{13}C^{16}O$ absorption bands and quantitative photochemical modeling as that by Lyons and coworkers[25], it is not possible to predict how exactly the current observation could affect the modeling outcomes, while we believe that consideration of the isotope dependent branching ratios on a state by state basis could improve any future quantitative photochemical models based on the CO self-shielding effect.

In summary, we have demonstrated a strong isotope effect in the VUV photodissociation of CO by measuring the branching ratios of $^{13}C^{16}O$ in the wavelength range 92.85–94.00 nm. The measurements showed that the branching ratio can be strongly altered through the substitution of $^{12}C$ by $^{13}C$. This isotope effect also depends on the specific rovibronic quantum state of CO that is being excited. We employed a simple diabatic coupling model to explain this strong isotope effect, which could potentially work for any other molecules which undergo indirect predissociation process into multiple channels, for example $N_2$[34]. This strong quantum state dependent isotope effect of the photodissociation branching ratio could potentially improve the current photo-chemical models, thus need to be considered on a state by state basis in any future quantitative modeling.

## Methods
**Photodissociation branching ratio measurement.** The experimental measure-ments in the current study were performed on the recently constructed mini-time-slice velocity-map ion imaging setup in our lab[37], together with a tunable VUV laser radiation source generated by the two-photon resonance-enhanced four-wave mixing scheme. A schematic diagram of the experiment is shown in Fig. 1. Briefly, a supersonic molecular beam of pure CO ($^{12}C^{16}O$, 99.9%; $^{13}C^{16}O$, $^{13}C$ = 99%, $^{18}O$ < 5%) generated by a pulsed valve (PV) operating at 10 Hz intersects with a VUV laser beam perpendicularly in the photodissociation and photoionization region. CO absorbs a single sum-frequency VUV photon ($2\omega_1+\omega_2$) and is excited to a specific rovibronic state and then undergoes predissociation through various pathways to form the three dissociation channels (as shown in Fig. 1b). Due to conservations of the total energy and linear momentum of the diatomic system, the following two equations can be deduced:

$$E_{tot} = D_o(CO) + E(C) + E(O) + E_{TKER} \quad (4)$$

$$E_{TKER} = (1 + M_C/M_O) \times E_C \quad (5)$$

$E_{tot}$ is the term energy of the excited state with respect to the ground rovibronic level of CO; $D_o(CO) = 11.09$ eV is the bond dissociation energy[47]; $E(C)$ and $E(O)$ are the internal energies of the C and O atoms respectively, with $E(C(^3P)) = E(O(^3P)) = 0$ eV, $E(C(^1D)) = 1.2637$ eV, $E(C(^1D)) = 1.9674$ eV; $E_{TKER}$ is the total kinetic energy release (TKER); $E_C$ is the kinetic energy of the C atoms; $M_C$ and $M_O$ are the masses of C and O atoms respectively. Since the three dissociation channels as shown in Eqs. (1–3) have atomic fragments with different internal energies, the resulting C atoms will have different kinetic energies (or different velocities). All the C atoms resulting from the same channel form a sphere in space, i.e. Newton sphere, the diameter of which is proportional to the velocities of the C atoms. Due to the existence of three energetically available channels, photodissociation of CO

usually forms three Newton spheres of different diameters, as shown in Fig. 1c. When the centers of the Newton spheres arrive at the MCP detector (Tectra, 50-D-L-P-FV, diameter of 50 mm), a 80 ns negative high voltage pulse is applied to the detector, thus only the central slices of the Newton spheres are detected. The thus formed images on the phosphor screen are captured by a CCD camera (LaVision, Imagery Pro Plus 4M), and processed through a computer program (Davis 8 Application Module), as shown in Fig. 1a.

According to eqs. (4) and (5), TKER spectra can be obtained by integrating the raw TSVMI images. Based on the fact that the diameters of the rings in the images are directly proportional to the C atom velocities, we can assign the three peaks observed in the TKER spectra to each of the three dissociation channels easily. We calculate the underlying areas of the three peaks, and then calibrate them with the different photoionization cross sections of $C(^3P)$ and $C(^1D)$ to obtain the branching ratios of the three dissociation channels as listed in Table 1. The photoionization cross section of $^{12}C(^3P)$ was measured to be ~16 Mb with an uncertainty of ~30%[38], and that of $^{12}C(^1D)$ was calculated to be ~27 Mb in the VUV range of the present study with no associated uncertainties reported[39], thus this could be the biggest potential uncertainty source of the current photodissociation branching ratio measurements. Since there are no relevant results on the photoionization cross sections of $^{13}C$, here we used the photoionization cross sections of $^{12}C$ to also calibrate the TKER spectra of $^{13}C^{16}O$.

**Tunable VUV source.** The tunable VUV laser radiation source used in the current study is generated by using the two-photon resonance-enhanced four-wave mixing scheme as shown in Fig. 1d. The second (green, 532 nm) and third (purple, 355 nm) harmonic outputs of a single Nd:YAG laser (Quanta-Ray, Pro-270-10E) are used to pump two dye lasers (Sirah, Cobra-Stretch) at the same time. The output from the one pumped by the third harmonic is frequency doubled by going through a fre-quency doubling unit (FDU), which gives out the UV laser beam $\omega_1$. $\omega_1$ is fixed at 222.568 nm ($\omega_1$) which is in resonance with the two-photon transition of Xe: $(5p)^5$ $(^2P_{1/2})6p^2[1/2](J=0) \leftarrow (5p)^6\ ^1S_0$ at 89,860.018 cm$^{-1}$. The dye laser pumped by the second harmonic gives out the tunable visible laser beam $\omega_2$, which was scanned from 560 to 610 nm in the present study. The UV ($\omega_1$) and visible ($\omega_2$) laser beams were combined through a dichroic mirror (DM) and focused into a T-shape channel by two plano-convex lens both with f = 600 mm. The Xe gas is pulsed into the T-shape channel when the two laser beams arrive, and the four-wave mixing process takes place to generate the tunable VUV laser radiation, which has the photon energy of $2\omega_1+\omega_2$. The time duration of the VUV laser pulse is 5–10 ns, and the predissociation life times of CO are usually in the picosecond range, thus in the present study a single VUV laser pulse is enough to first dissociate CO molecules and then ionize the C atoms for detection in the same laser pulse through a direct photoionization process. The VUV intensity generated this way is generally weak, no saturation of the pho-toionization has been noticed, thus the absolute VUV intensity should have no effect on the relative branching ratio measurements.

## Data availability
All the data presented in the present study are available from the authors on request for academic use.

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

## Acknowledgements

This work is supported by the National Natural Science Foundation of China (Grant No. 21803072), the Program for Young Outstanding Scientists of Institute of Chemistry, Chinese Academy of Science (ICCAS), and Beijing National Laboratory for Molecular Sciences (BNLMS). We thank Prof. Yang Pan (National Synchrotron Radiation Laboratory, University of Science and Technology of China) for instrumentation supports.

## Author contributions

P.J. and X.P.C. collected and analyzed the data. H.G., M.C. and Q.H.Z. conceived and supervised the project. H.G. wrote the manuscript and all authors have read and approved its final form.

## Additional information

**Competing interests:** The authors declare no competing interests.

