## [Peer Review File · Nature Communications]

Reviewers' comments:

Reviewer #1 (Remarks to the Author):

The authors present measured branching ratios for CO dissociation at VUV wavelengths (92.8 to 94.0 nm). They demonstrate preferential formation of $^{13}\text{C}(1\text{D})$ for a couple of the bands in this region, and argue that such a preference for ^{13}C formation may be important for understanding the difference in the $^{12}\text{C}/^{13}\text{C}$ ratio for the Sun versus planetary materials. They apply a simple diabatic model to explain the preferential formation of $^{13}\text{C}(1\text{D})$ in their experiments. They then expand their diabatic arguments to O isotopes branching ratios in CO dissociation.

The measurements presented seem to be of good quality. However, they are of a small subset of CO bands in a moderately congested region of the CO spectrum. Since $\text{C}(1\text{D})$ formation is energetically allowed up to ~ 100 nm, it would make vastly more sense to present experimental results for isotopic branching ratios for some of the bands between 94 and 100 nm. It is impossible to draw conclusions about the importance of the 1.2 nm region they have studied for the simple reason that it contributes only a small fraction to the total rate of CO photodissociation.

Second, the rather extensive discussion of O isotopes fractionation, and the experiments of the Thieme group, is completely unsupported by the data presented. If the authors wish to make a meaningful contribution to the understanding of O isotope fractionation in CO dissociation, then they should perform experiments using C^{16}O , C^{17}O , and C^{18}O . The discussion of O isotopes they have presented is merely conjecture, and does not acknowledge the role of other factors such as linewidth for the short wavelength CO bands. Line overlap has an important effect on O isotope fractionation.

For the above two reasons I cannot recommend this paper for publication in its present form.

Additional comments:

1. The excited state $\text{C}'1\Sigma^+(\nu'=7)$ is referred to repeatedly throughout the text. Is this a new state assignment (and if so please give the citation)? Usually this state is simply the $1\Sigma^+(\nu'=2)$ state.

2. The potential energy curves in Figure 1b are not really useful in this cartoon format. A more accurate and properly labeled set of PECs is needed.

Reviewer #2 (Remarks to the Author):

I recommend the paper for publication with only minor revisions.

The new results – concerning the strong isotope-dependence of photodissociation branching ratios in carbon monoxide – are relevant to ongoing studies of predissociation mechanisms in CO. In addition to the intrinsic merit of a more detailed understanding of the interactions of high-lying electronic states in CO, there are important astrophysical applications that rely on a quantitative understanding of predissociation in CO and its isotopologues. This paper highlights the previously under-appreciated role of branching ratio differences among the CO isotopologues as a possible explanation for the strong fractionation signatures of carbon and oxygen isotopes in the solar system. The qualitative analysis presented in the paper – a coupled-channel model of diabatic states leading to indirect predissociation – is a sensible explanation for the trends seen in the experimental results. The experimental methods are state-of-the-art and the level of experimental detail presented is appropriate.

Below I have indicated some suggested minor revisions, but there is no need for further review.

1. page 1, title: please replace “on the vacuum...” with “in the vacuum...”
2. page 2, line 41: in the introductory discussion, the term “To test the self-shielding models” will be confusing to anyone who is not intimately familiar with the works being cited. The authors should define this term – a single sentence would help to clarify the introductory remarks.
3. page 7, line 200: the phrase “the substitution of ^{12}C by ^{13}C must have dramatically changed the relative coupling strengths among different electronic states” should be re-phrased. As the authors later mention, the coupling between electronic states is independent of isotope. It is the coupling between individual rovibronic levels that depends on isotope (not the coupling between electronic states) – because of the energy shifts that the authors describe.

4. Uncertainty estimates are included in the results presented in Table 1, however there is no discussion whatsoever about uncertainties in the text. The authors should include a brief description of the uncertainty levels in the text itself and how they came to determine those uncertainties.

5. The grammar in the manuscript needs a careful review by the authors and/or the editors of the journal. Multiple awkward phrasings detract from the quality of the science being presented.

Regards,

Glenn Stark

Reviewer #3 (Remarks to the Author):

The authors report on a compelling experimental study that characterizes a strong and selective isotopic effect of photodissociation branching ratios of carbon monoxide in the VUV absorption range. They use VUV excitation generated by two photon resonant enhanced four wave mixing coupled to a time-slice velocity-map ion imaging apparatus (TSVMI) to measure the branching ratios in the Rydberg $4p(2)$, $5p(0)$ and $5s(0)$ complex region for three indirect predissociative pathways for the two isotopomers : $^{12}\text{C}^{16}\text{O}$ or $^{13}\text{C}^{16}\text{O}$: one leading to the atoms in their ground triplet ($3P$)states, one leading to $\text{C}(1D) + \text{O}(3P)$ and one leading to $\text{C}(3P)$ and $\text{O}(1D)$. They report on a significant isotopic effect for the predissociation channel leading to $\text{C}(1D) + \text{O}(3P)$ for the absorption bands $1\Pi(v'=2)$ and $C' 1\Sigma+(v'=7)$. For these bands, $^{12}\text{C}^{16}\text{O}$ primarily dissociates in the lowest channel while $^{13}\text{C}^{16}\text{O}$ dissociates to the next channel producing to $\text{C}(1D)$ atoms. Significant variations of the branching ratios are also reported for the other absorption bands in this spectral region. To explain the strong experimental effect, the authors adopt the explanation proposed in ref 34 : due to the mass effect, different vibronic states are excited in the two isotopomers : in the case when the predissociation process is indirect and involves an electronic coupling between a Rydberg and a valence state, the ladder of rovibrational states in the two electronic states are shifted by the change in reduced mass, more in the case of the shallow valence state, which leads to different efficiencies of the electronic coupling for the two isotopomers due to the fact that accidental resonances occur at slightly different energies in the two isotopomers. As noted already in ref 34 and reinstated by the authors, this mechanism is general. In this sense, I recommend that the authors tone down the qualification 'new', 'novel', 'for the first time', that they use in several places in the manuscript. It is very important to document experimental the ubiquity of the mechanism proposed in ref 34 for N_2 in case of indirect predissociation for other diatomic molecules but it can no longer be qualified to be novel. The authors might want to consider to also refer to more recent

work of the authors of ref 34, for example J. S. Ajay, K. G. Komarova, F. Remacle and R. D. Levine, Proceedings of the National Academy of Sciences, 2018, 115, 5890.

The paper is clearly written and reports on important experimental results that will stimulate more theoretical work. I recommend its publication in nature communication.

First, we really want to thank all the three reviewers for carefully reading our manuscript, and give us many valuable and important suggestions in such short time. We have carefully read all the comments, and revised the manuscript accordingly. The detailed replies point-by-point are listed as follows:

Reviewer #1 (Remarks to the Author):

The authors present measured branching ratios for CO dissociation at VUV wavelengths (92.8 to 94.0 nm). They demonstrate preferential formation of $^{13}\text{C}(1\text{D})$ for a couple of the bands in this region, and argue that such a preference for ^{13}C formation may be important for understanding the difference in the $^{12}\text{C}/^{13}\text{C}$ ratio for the Sun versus planetary materials. They apply a simple diabatic model to explain the preferential formation of $^{13}\text{C}(1\text{D})$ in their experiments. They then expand their diabatic arguments to O isotopes branching ratios in CO dissociation.

Our reply: The reviewer is right about the logic that we are using in the manuscript, that we measured several typical absorption bands of $^{13}\text{C}^{16}\text{O}$ in a small energy range, and found that the branching ratios are very different from that of $^{12}\text{C}^{16}\text{O}$, then we argue that the underlying mechanism can be general for other bands of $^{13}\text{C}^{16}\text{O}$ that are not measured yet, and also for $^{12}\text{C}^{17}\text{O}$ and $^{12}\text{C}^{18}\text{O}$. We believe that this logic generally makes sense, which I hope the reviewer can also agree with.

We clearly noted that this isotope effect on the photodissociation branching ratio is very random, it could produce more $\text{C}(1\text{D})$, it could also produce less $\text{C}(1\text{D})$, depending on the specific absorption band, this has been clearly shown in the manuscript. We are also clear that the overall effect of this isotope effect cannot be quantified without measuring all the relevant absorption bands and quantitative modeling, thus we did not make any conclusions anywhere in the manuscript about if this observation will prefer or not prefer the overall $^{13}\text{C}(1\text{D})$ formation. The only conclusions that we have emphasized are: isotope substitution can strongly affect the photodissociation branching ratios, and this isotope effect depends on specific absorption band, and it need be considered in photochemical models on a state-by-state basis. To this point, we think the amount of experiment that we present in the manuscript is enough to support the conclusions that we made.

The measurements presented seem to be of good quality. However, they are of a small subset of CO bands in a moderately congested region of the CO spectrum. Since $\text{C}(1\text{D})$ formation is energetically allowed up to ~ 100 nm, it would make vastly more sense to present experimental results for isotopic branching ratios for some of the bands between 94 and 100 nm. It is impossible to draw conclusions about the importance of the 1.2 nm region they have studied for the simple reason that it contributes only a small fraction to the total rate of CO photodissociation.

Our reply: The reviewer is correct that we have only measured a very small range for $^{13}\text{C}^{16}\text{O}$, while as I said above, we are not trying to make any “real final conclusions” about the possible impacts on the astrophysical models, because to do this, we need finish measuring all the ~ 30 absorption bands in the range 94-100 nm, and also higher energy range 91.1-92.8 nm, and then perform a quantitative modeling considering the self-shielding effect and also other isotopic

effects, for example the isotope dependent dissociation rates, like that has been done by Lyons in Ref. 25. This is very large amount of experimental and modeling work, which we think is beyond the scope of a single communication paper like this.

We are now on the way to measure the branching ratios of all the strong absorption bands of $^{13}\text{C}^{16}\text{O}$, $^{12}\text{C}^{17}\text{O}$ and $^{12}\text{C}^{18}\text{O}$ step by step, while this will take very long time. Since the submission of this manuscript, we have now measured the branching ratios of $^{13}\text{C}^{16}\text{O}$ for most of the strong absorption bands in the range 94-100nm (not completed yet), and have obtained some preliminary data. The branching ratios for higher energy range 91.1-92.8 nm have also been planned, but not measured yet. The current preliminary data in the range 94-100nm we obtained showed that several absorption bands have enhanced $\text{C}(^1\text{D})$ productions by 20%-30%, like $\text{W}(3s\sigma)^1\Pi(v'=2)$, $\text{L}(4p\pi)^1\Pi(v'=0)$ and $\text{K}(4p\sigma)^1\Pi(v'=0)$; there are also bands with reduced $\text{C}(^1\text{D})$ productions, like $\text{W}(3s\sigma)^1\Pi(v'=1)$, by about 15%-20%. Even though we have not completed the measurements in the 94-100nm range so far yet, this observation already confirms the results and conclusions we made in the present manuscript. **We hope that the reviewer can agree with us on not putting these new data which are still very preliminary into the present manuscript, because it will make our data set very fragmented, and result in major re-organization of the manuscript, and to our opinion this effort will not make the present manuscript stronger, as it is still not enough to make any real conclusions about the overall production rates of $^{13}\text{C}(^1\text{D})$ without measurements in the range 91.1-92.8 nm and any quantitative modeling by considering the self-shielding effect and the dissociation cross sections of all these absorption bands. We would prefer to presenting the new data in 94-100 nm wavelength range in subsequent publications until we finish all the data analysis.**

To make things more clear, we have added one more sentence in the second paragraph of the section "Possible impacts on photochemical models" as "Although without systematic branching ratio measurements like that in the present study for all the relevant $^{13}\text{C}^{16}\text{O}$ absorption bands and quantitative photochemical modeling as that by Lyons and coworkers²⁵, it is not possible to predict how the current observation could affect the modeling outcomes". We have also changed the phrase "definitely necessary" to "could improve", to tone down the possible importance of the branching ratio measurement.

Second, the rather extensive discussion of O isotopes fractionation, and the experiments of the Thiemens group, is completely unsupported by the data presented. If the authors wish to make a meaningful contribution to the understanding O isotope fractionation in CO dissociation, then they should perform experiments using C^{16}O , C^{17}O , and C^{18}O . The discussion of O isotopes they have presented is merely conjecture, and does not acknowledge the role of other factors such as linewidth for the short wavelength CO bands. Line overlap has an important effect on O isotope fractionation.

Our reply: In fact, we are initially motivated by the O isotopic fractionation effects in the Solar system and wanted to measure the branching ratios for $^{12}\text{C}^{18}\text{O}$ at the beginning, while the sample of enriched $^{12}\text{C}^{18}\text{O}$ is super expensive ($^{12}\text{C}^{17}\text{O}$ is even more expensive) and not easy to obtain in China, then we decide to measure the branching ratios of $^{13}\text{C}^{16}\text{O}$ which is much cheaper, to see if there are any isotopic effects. It is also because that we think the C and O problems associated with the photodissociation of CO are closely related to each other, and the current measurements

on $^{13}\text{C}^{16}\text{O}$ could strongly imply that $^{12}\text{C}^{17}\text{O}$ and $^{12}\text{C}^{18}\text{O}$ can also have different branching ratios compared with that of $^{12}\text{C}^{16}\text{O}$. We are further motivated by this experiment, and have already ordered a bottle of $^{12}\text{C}^{18}\text{O}$ from USA which is already on the way. In the introduction section, we mentioned both C and O for generally emphasizing the importance of carefully measuring the isotope dependent dissociation branching ratios, which is also the initial motivation of the whole project which is being supported by Chinese National Science Foundation.

We agree with the reviewer that self-shielding effect still plays the major role on O isotope fractionation process, and other factors can affect the results, and **the idea we want to present here is that the branching ratio may be also one of these other factors**, and this is our motivation for measuring the branching ratio of $^{12}\text{C}^{17}\text{O}$ and $^{12}\text{C}^{18}\text{O}$ in the next step. To make this clear, we have rewritten the first paragraph of the section “Possible impacts on the photochemical models”, please refer to the revised manuscript for details. We have emphasized the importance of the self-shielding effect and other isotopic effects as requested by the reviewer, and hope that this can be accepted by the reviewer.

For the above two reasons I cannot recommend this paper for publication in its present form.

Additional comments:

1. The excited state $\text{C}'1\Sigma^+(\nu'=7)$ is referred to repeatedly throughout the text. Is this a new state assignment (and if so please give the citation)? Usually this state is simply the $1\Sigma^+(\nu'=2)$ state.

Our reply: This state was assigned as $\text{C}'1\Sigma^+(\nu'=7)$ in Ref. 8. We have added a footnote under Table I to make it clear.

2. The potential energy curves in Figure 1b are not really useful in this cartoon format. A more accurate and properly labeled set of PECs is needed.

Our reply: There are two reasons that we use a simple cartoon picture for illustrating the dissociation mechanisms here. First, accurate PECs for CO which include the information of mutual coupling strengths among different electronic states in such high energy region are still not available, thus it is not really possible to specifically label all the interacting potential curves; second, Nature Communications are dedicated to very broad scopes of readers from different areas, a simple cartoon picture which is illustrative for readers who are not familiar with molecular photodissociation would be very helpful. I hope the reviewer can agree on this.

Reviewer #2 (Remarks to the Author):

I recommend the paper for publication with only minor revisions.

The new results – concerning the strong isotope-dependence of photodissociation branching ratios in carbon monoxide – are relevant to ongoing studies of predissociation mechanisms in CO. In addition to the intrinsic merit of a more detailed understanding of the interactions of high-lying

electronic states in CO, there are important astrophysical applications that rely on a quantitative understanding of predissociation in CO and its isotopologues. This paper highlights the previously under-appreciated role of branching ratio differences among the CO isotopologues as a possible explanation for the strong fractionation signatures of carbon and oxygen isotopes in the solar system. The qualitative analysis presented in the paper – a coupled-channel model of diabatic states leading to indirect predissociation – is a sensible explanation for the trends seen in the experimental results. The experimental methods are state-of-the-art and the level of experimental detail presented is appropriate.

Our reply: Thank you for the very positive comment on our work!

Below I have indicated some suggested minor revisions, but there is no need for further review.

1. page 1, title: please replace “on the vacuum...” with “in the vacuum...”

Our reply: Fixed.

2. page 2, line 41: in the introductory discussion, the term “To test the self-shielding models” will be confusing to anyone who is not intimately familiar with the works being cited. The authors should define this term – a single sentence would help to clarify the introductory remarks.

Our reply: We have added the follow two sentences to the introduction section: Self-shielding process happens when a light beam with broad spectral distribution penetrates a sample of mixed gaseous molecules which have slightly different absorption line positions. Due to the different column densities of the gaseous components, which result in different attenuation speeds according to Beer’s law, the relative amount of light absorption for different molecular species will change along the light propagation direction.

3. page 7, line 200: the phrase “the substitution of ^{12}C by ^{13}C must have dramatically changed the relative coupling strengths among different electronic states” should be re-phrased. As the authors later mention, the coupling between electronic states is independent of isotope. It is the coupling between individual rovibronic levels that depends on isotope (not the coupling between electronic states) – because of the energy shifts that the authors describe.

Our reply: We have changed “among different electronic states” to “among different rovibronic states”.

4. Uncertainty estimates are included in the results presented in Table 1, however there is no discussion whatsoever about uncertainties in the text. The authors should include a brief description of the uncertainty levels in the text itself and how they came to determine those uncertainties.

Our reply: A footnote for describing the error bars is added to Table I.

5. The grammar in the manuscript needs a careful review by the authors and/or the editors of the journal. Multiple awkward phrasings detract from the quality of the science being presented.

Our reply: We have reviewed our manuscript for many times, and have tried our best to make all grammars correct before we submit it. We will do this again.

Regards,
Glenn Stark

Reviewer #3 (Remarks to the Author):

The authors report on a compelling experimental study that characterizes a strong and selective isotopic effect of photodissociation branching ratios of carbon monoxide in the VUV absorption range. They use VUV excitation generated by two photon resonant enhanced four wave mixing coupled to a time-slice velocity-map ion imaging apparatus (TSVMI) to measure the branching ratios in the Rydberg $4p(2)$, $5p(0)$ and $5s(0)$ complex region for three indirect predissociative pathways for the two isotopomers : $^{12}\text{C}^{16}\text{O}$ or $^{13}\text{C}^{16}\text{O}$: one leading to the atoms in their ground triplet (3P) states, one leading to $\text{C}(1\text{D}) + \text{O}(3\text{P})$ and one leading to $\text{C}(3\text{P})$ and $\text{O}(1\text{D})$. They report on a significant isotopic effect for the predissociation channel leading to $\text{C}(1\text{D}) + \text{O}(3\text{P})$ for the absorption bands $1\Pi(v'=2)$ and $\text{C}' 1\Sigma+(v'=7)$. For these bands, $^{12}\text{C}^{16}\text{O}$ primarily dissociates in the lowest channel while $^{13}\text{C}^{16}\text{O}$ dissociates to the next channel producing to $\text{C}(1\text{D})$ atoms. Significant variations of the branching ratios are also reported for the other absorption bands in this spectral region. To explain the strong experimental effect, the authors adopt the explanation proposed in ref 34 : due to the mass effect, different vibronic states are excited in the two isotopomers : in the case when the predissociation process is indirect and involves an electronic coupling between a Rydberg and a valence state, the ladder of rovibrational states in the two electronic states are shifted by the change in reduced mass, more in the case of the shallow valence state, which leads to different efficiencies of the electronic coupling for the two isotopomers due to the fact that accidental resonances occur at slightly different energies in the two isotopomers. As noted already in ref 34 and reinstated by the authors, this mechanism is general. In this sense, I recommend that the authors tone down the qualification 'new', 'novel', 'for the first time', that they use in several places in the manuscript. It is very important to document experimental the ubiquity of the mechanism proposed in ref 34 for N_2 in case of indirect predissociation for other diatomic molecules but it can no longer be qualified to be novel. The authors might want to consider to also refer to more recent work of the authors of ref 34, for example J. S. Ajay, K. G. Komarova, F. Remacle and R. D. Levine, Proceedings of the National Academy of Sciences, 2018, 115, 5890.

Our reply: Thank you for the positive comments on our work. As requested, we have deleted all the phrases like 'new', 'novel', 'for the first time' in the manuscript. We have also added the most recent reference as recommended by the reviewer.

The paper is clearly written and reports on important experimental results that will stimulate more theoretical work. I recommend its publication in nature communication.

Reviewers' comments:

Reviewer #1 (Remarks to the Author):

I thank the authors for the additions to the text based on referee comments. I recommend the paper for publication, but I do not do so with enthusiasm, and only after several minor but important revisions are made. My lack of enthusiasm is for precisely the reason I gave in my first review: the results presented are inadequate to draw conclusions about their broader importance outside of VUV spectroscopy. They do however point out the need for more spectroscopic work.

Minor revisions needed:

1. line 50-51: the 'debate' discussed has long since been settled. Self-shielding is, without question, the source of the massive fractionation observed in the Chakraborty experiments. Can an isotopic dependence in quantum yields (long-wavelength E and C bands) affect the relative amount of fractionation for different isotopes (i.e., ^{17}O and ^{18}O)? Yes. Could the branching ratio mechanism presented here also affect the distribution of isotope fractionation? Yes, although only at wavelengths ~ 100 nm or less. The authors need to describe this 'debate' more clearly and with greater accuracy.

2. line 97: There have been a great many laboratory SO_2 photolysis experiments performed in the C-X system (~ 180 - 220 nm). Isotopic measurements of sulfur product from these experiments have only shown clear evidence for self-shielding by $^{32}\text{SO}_2$. Perhaps at shorter wavelengths the branching ratio mechanism discussed here could be important.

3. line 247: grammar - remove 'While'

4. lines 300-315: In the new text the authors need to add the important point that $\text{O}(1\text{D})$ production only occurs at wavelengths < 95 nm. Therefore production of $^{17}\text{O}(1\text{D})$ and $^{18}\text{O}(1\text{D})$, and a possible modification of their branching ratios relative to $^{16}\text{O}(1\text{D})$ is only possible for < 95 nm photons. That means that most of the Chakraborty O isotope results cannot be explained by the branching ratio mechanism presented here. It is possible that the $^{12}\text{C}(1\text{D})$ branch will affect $^{17}\text{O}(3\text{P})$ and $^{18}\text{O}(3\text{P})$ branching ratios, which, if so, would extend the process to ~ 100 nm photons. But this still would not account for the Chakraborty results at 105 and 107 nm (the important E1 and E0 bands). The authors must clearly state this limitation to the branching ratio mechanism for O isotopes.

5. line 320: change 'photochemical model' to 'photochemical timescale model'

6. line 347: see above about SO₂

7. lines 388-389: Please describe the source and magnitude of the uncertainty in the C(3P) and C(1D) photoionization cross sections. Does the laser radiation field contribute significantly to the uncertainty on photoionization rates, or is it essentially perfectly known?

Reviewer #1 (Remarks to the Author):

I thank the authors for the additions to the text based on referee comments. I recommend the paper for publication, but I do not do so with enthusiasm, and only after several minor but important revisions are made. My lack of enthusiasm is for precisely the reason I gave in my first review: the results presented are inadequate to draw conclusions about their broader importance outside of VUV spectroscopy. They do however point out the need for more spectroscopic work.

Our reply: Thank the referee for agreeing with most of our revisions. We will definitely continue our effort on collecting the branching ratios for all the relevant absorption bands of $^{13}\text{C}^{16}\text{O}$, and also for $^{12}\text{C}^{17}\text{O}$ and $^{12}\text{C}^{18}\text{O}$, and hope finally we can make substantial progress toward understanding the massive isotopic fractionation effects observed in the solar system quantitatively.

Minor revisions needed:

1. line 50-51: the 'debate' discussed has long since been settled. Self-shielding is, without question, the source of the massive fractionation observed in the Chakraborty experiments. Can an isotopic dependence in quantum yields (long-wavelength E and C bands) affect the relative amount of fractionation for different isotopes (i.e., 17O and 18O)? Yes. Could the branching ratio mechanism presented here also affect the distribution of isotope fractionation? Yes, although only at wavelengths ~ 100 nm or less. The authors need to describe this 'debate' more clearly and with greater accuracy.

Our reply: We agree with the referee's comments. We have change the sentence to "Now it is generally agreed that self-shielding effect is the main reason of the massive isotope fractionation observed in the above experiment and also that in the solar system, while Thiemens' experiment..."

2. line 97: There have been a great many laboratory SO_2 photolysis experiments performed in the C-X system (~ 180 - 220 nm). Isotopic measurements of sulfur product from these experiments have only shown clear evidence for self-shielding by $^{32}\text{SO}_2$. Perhaps at shorter wavelengths the branching ratio mechanism discussed here could be important.

Our reply: Thank you for pointing this out to us. Actually, SO_2 and its sulfur isotopomers are our next target molecules for measuring the product quantum state distributions, since according to Ref. 47, the quantum state distributions of SO_2 do depend on the sulfur isotope substitution, while this might have no effect on the sulfur isotope fractionation process. To not to make any misunderstandings, we will just avoid mentioning SO_2 here and also in Comment 6.

3. line 247: grammar - remove 'While'

Our reply: Fixed.

4. lines 300-315: In the new text the authors need to add the important point that O(1D) production only occurs at wavelengths < 95 nm. Therefore production of 17O(1D) and 18O(1D), and a possible modification of their branching ratios relative to 16O(1D) is only possible for < 95 nm photons. That means that most of the Chakraborty O isotope results cannot be explained by the branching ratio mechanism presented here. It is possible that the 12C(1D) branch will affect 17O(3P) and 18O(3P) branching ratios, which, if so, would extend the process to ~ 100 nm photons. But this still would not account for the Chakraborty results at 105 and 107 nm (the important E1 and E0 bands). The authors must clearly state this limitation to the branching ratio mechanism for O isotopes.

Our reply: We totally agree with this. We have added a sentence in this section “Since CO photodissociation can only generate excited O atoms [O(¹D)] at VUV wavelength shorter than 94.94 nm (see reaction (3)), thus the present isotope effect observed for the branching ratio should not affect any absorption bands at longer wavelength in Thiemens’ experiment.”

5. line 320: change 'photochemical model' to 'photochemical timescale model'

Our reply: Fixed

6. line 347: see above about SO₂

Our reply: See Comment 2.

7. lines 388-389: Please describe the source and magnitude of the uncertainty in the C(3P) and C(1D) photoionization cross sections. Does the laser radiation field contribute significantly to the uncertainty on photoionization rates, or is it essentially perfectly known?

Our reply: We have added several sentences in the Methods section to clarify this. For C(³P), they reported the uncertainty to be about 30%, while for C(¹D), no uncertainty of the photoionization cross section was reported. This will increase the uncertainties of the branching ratios in Table I, while it does not affect the comparison between ¹²C¹⁶O and ¹³C¹⁶O.

We do not perfectly know the intensity of the sum-frequency VUV, and it varies from day to day. While the VUV intensity is generally weak, and no saturation of the photoionization was noticed, thus it should not affect the relative branching ratio measurements. This has also been confirmed by the fact that we obtained nearly the same value on different days. We have added several sentences at the end of the Methods section.

Reviewer #1 (Remarks to the Author):

The authors have adequately addressed the points I raised. I recommend the manuscript for publication.

Reply: Thanks!